biochemistry/biotechnology/microbiology

microbial surfactants, biosurfactant-producing bacteria, petroleum hydrocarbon bioremediation, crude oil biodegradation, emulsification index, production optimization

**Authors for correspondence:**
Ferdausi Ali
e-mail: seema@cu.ac.bd
Tanim Jabid Hossain
e-mail: tanim.bmb@gmail.com

# Production optimization, stability and oil emulsifying potential of biosurfactants from selected bacteria isolated from oil-contaminated sites

Ferdausi Ali[1], Sharup Das[1], Tanim Jabid Hossain[2], Sumaiya Islam Chowdhury[2], Subrina Akter Zedny[1,2], Tuhin Das[1], Mohammad Nazmul Ahmed Chowdhury[2] and Mohammad Seraj Uddin[1]

[1]Department of Microbiology, and [2]Department of Biochemistry and Molecular Biology, University of Chittagong, Chattogram 4331, Bangladesh

SD, 0000-0002-3151-0885; TJH, 0000-0002-0978-2657;
SIC, 0000-0003-1011-4995; SAZ, 0000-0002-6835-4353;
TD, 0000-0003-4360-5619; MNAC, 0000-0001-7355-7924;
MSU, 0000-0003-0244-5698

Oil pollution is of increasing concern for environmental safety and the use of microbial surfactants in oil remediation has become inevitable for their efficacy and ecofriendly nature. In this work, biosurfactants of bacteria isolated from oil-contaminated soil have been characterized. Four potent biosurfactant-producing strains (SD4, SD11, SD12 and SD13) were selected from 27 isolates based on drop collapse assay and emulsification index, and identified as species belonging to *Bacillus*, *Burkholderia*, *Providencia* and *Klebsiella*, revealed from their 16S rRNA gene-based analysis. Detailed morphological and biochemical characteristics of each selected isolate were determined. Their growth conditions for maximum biosurfactant production were optimized and found quite similar among the four isolates with a pH of 3.0 and temperature 37°C after 6 or 7 days of growth on kerosene. The biosurfactants of SD4, SD11 and SD12 appeared to be glycolipids and that of SD13 a lipopeptide. Emulsification activity of most of the biosurfactants was stable at low and high temperatures (4–100°C), a wide range of pH (2–10) and salt concentrations (2–7% NaCl). Each biosurfactant showed

antimicrobial activity against two or more pathogenic bacteria. The biosurfactants were well-capable of emulsifying kerosene, diesel and soya bean, and could efficiently degrade diesel.

# 1. Introduction

Petroleum-based fuels have been a major concern for life and environment especially in industrialized and developing countries [1]. Over the years, numerous natural and anthropogenic incidents have led to an enormous release of petroleum oil into nature thus posing a serious threat to the quality and sustainability of ecosystems [2,3]. Petroleum oil contains many aromatic toxic compounds such as benzene, ethylbenzene, toluene, xylene, etc. that are harmful for most life forms [4–7]. In addition to causing physical damage to habitats, the toxic ingredients of petroleum make mutagenic and carcinogenic changes to people [8]. Exposure to benzene and benzopyrene, for example, was found associated with an increased risk of leukaemia and lung cancer, respectively [9,10]. However, microbial populations particularly some bacteria and fungi manage to thrive on these rather harmful aromatic pollutants [11]. Regardless of the pollutants' high toxicity and hydrophobicity, bacterial species including those of the genera *Pseudomonas*, *Bacillus*, *Streptomyces*, *Stenotrophomonas*, etc. have been found inhabiting such petroleum-rich niches [12,13]. One of the key properties that allow these microbes to endure polluted environments is their ability to uptake petroleum hydrocarbons and facilitate their degradation by the production of a group of surface-active agents known as biosurfactants [14]. The biosurfactants are excreted from microbial cells or produced at the cell surface and include a broad range of chemical structures with diverse surface properties. They can be low molecular weight biomolecules that are generally glycolipids such as trehalose lipids, sophorolipids and rhamnolipids or lipopeptides such as surfactin, gramicidin S and polymyxin, or high molecular weight compounds such as polysaccharides, proteins, lipopolysaccharides, lipoproteins or complex mixtures of these biopolymers [15]. Biochemically, the microbial surfactants consist of both hydrophilic and hydrophobic moieties [16]. This amphipathic nature allows biosurfactants to partition at the interface between aqueous and hydrophobic phases, e.g. oil and water, or oil and rock interfaces, thus reducing the surface and interfacial tensions [17]. The biosurfactants, therefore, appear very effective in mobilization, increase of bioavailability and degradation of residual oil at a contaminated area [18]. Other important functions of biosurfactants include antimicrobial and antiviral activities, immunomodulation, enzyme inhibition, regulation of cell surface properties facilitating attachment to or detachment from surfaces, etc. [19,20].

A variety of synthetic or chemical surfactants are also available and used for the environmental bioremediation of petroleum hydrocarbons [18]. But microbial surfactants are of particular importance in this regard since they offer several advantages over their synthetic counterparts. For example, biosurfactants show better foaming capacity, selectivity and specific activity as compared to the synthetic surfactants [21]. Moreover, biosurfactants, due to their higher biodegradability, are less toxic than the chemical surfactants [22]. In addition, microbial surfactants are more stable and efficient over a wide range of environmental conditions, e.g. temperatures, pH and salinity [23]. Hence, biosurfactants are considered better candidates for environmental oil recovery processes and supposed to replace the synthetic surfactants.

Due to the importance of microbial surfactants, the present study has been carried out to isolate efficient biosurfactant-producing bacteria and characterize their secreted surfactants for potential application in hydrocarbon bioremediation. Hence, bacterial species isolated from oil-contaminated sites were screened for biosurfactant production and selected isolates were examined for optimum yields at various culture conditions. Morphological, biochemical and taxonomic characteristics of the isolates and preliminary characteristics of their surfactants have also been studied. Additionally, the degradation of diesel oil by the biosurfactants under laboratory conditions was evaluated.

# 2. Material and methods

## 2.1. Soil samples

The soil was collected from three different locations of Chittagong (see electronic supplementary material, figure S1) in April 2018. The top layer (0–15 cm) of the surface soil was collected using sterile spatula into

**Table 1.** Location and physico-chemical properties of the samples.

| location | GPS coordinates | soil temperature | soil pH | strains isolated |
|---|---|---|---|---|
| Shoraipara fuel station, Pahartali | 22.35586828 N, 91.78846921 E | 30°C | 8 | SD1, SD2, SD3, SD4 |
| engine filling station, Chittagong Railway Academy | 22.32053532 N, 91.78405199 E | 31°C | 8.5 | SD5, SD6, SD7 |
| engine washing station, Chittagong Railway Academy | 22.32338341 N, 91.78185608 E | 29°C | 9 | SD8, SD9, SD10 |
| main station, Chittagong Railway Academy | 22.32121629 N, 91.78552304 E | 32°C | 8 | SD11, SD12, SD13 |

sterile zip-locked bags and kept in an icebox during transportation to the laboratory. The physico-chemical properties of the soil, e.g. pH and the temperature, were measured at the collection sites (table 1).

## 2.2. Enrichment and isolation

One gram of each soil sample was dissolved in 99 ml of Mckeen medium containing 25 g glucose, 2.5 g monosodium glutamate, 3.0 g yeast extract, 1.0 g $MgSO_4 \cdot 7H_2O$, 1.0 g $K_2HPO_4$, 0.5 g KCl and 1.0 ml trace element solution (0.64 g $MnSO_4 \cdot 7H_2O$, 0.16 g $CuSO_4 \cdot 5H_2O$ and 0.015 g $FeSO_4 \cdot 7H_2O$ in 100 ml of distilled water) per 1 l distilled water [24]. After incubation at 37°C for 3 days at 150 r.p.m., 100 µl of the suspension was spread over Mckeen agar plates and incubated at 37°C. Single colonies from the plate were picked and repeatedly streaked on fresh plates until pure cultures appeared that were preserved as slant cultures.

## 2.3. Hydrocarbon overlay assay

Initial screening of the isolates for biosurfactant production was performed by hydrocarbon overlay assay as described by Hanano *et al.* [25]. One microliter of culture was spread over a McKeen agar plate coated with 100 µl of kerosene and incubated at 37°C for 7 days. Colony surrounded by an emulsified halo was considered positive for biosurfactant production.

## 2.4. Drop collapse assay

Drop collapse assay was carried out according to the description of [26] using cell-free supernatant prepared from the centrifugation of a 48 h culture at 5000 r.p.m. for 20 min at 4°C. A single drop of diesel oil was placed on a glass slide upon which one drop of the supernatant was dropped. After 1–2 min, the flattening property was recorded. If the drop collapsed the result was scored as positive while if it remained beaded the result was considered negative.

## 2.5. Blood agar assay

To perform blood agar assay, fresh cultures were streaked on blood agar plates (Himedia, India) containing 5–7% sheep blood. After incubation at 37°C for 48–72 h, the formation of a clear halo surrounding the colonies was scored as a positive result [27].

## 2.6. Determination of emulsification index

The emulsification index ($E_{24}$) was determined as previously reported [28]. Three microlitres of kerosene was added to the same amount of cell-free supernatant and vortexed for 2 min. After 24 h, the height of the stable emulsion layer was measured. Water was used as negative control. $E_{24}$ was defined as the percentage of the height of the emulsified layer divided by the total height of the liquid column:

$$E_{24} = \frac{\text{height of the emulsion layer}}{\text{total height}} \times 100\%.$$

## 2.7. PCR and sequencing of 16S rRNA gene

To amplify 16S rRNA gene sequences, cells from the stock culture were inoculated in nutrient broth containing 5.0 g peptone, 3.0 g yeast extract, 5.0 g NaCl in 1 l distilled water and incubated overnight at 37°C. The activated cultures were further grown in nutrient broth at 37°C overnight and their genomic DNA was extracted using a Maxwell 16 Blood DNA Purification Kit (Promega, Madison, WI, USA) according to the manufacturer's instructions. PCR was carried out with the genomic DNA using the primers 27F (5′-AGAGTTTGATCNTGGCTCAG-3′) and 1492R (5′-GCTTACCTTGTTACGACTT-3′). Sequencing of the purified PCR products was performed as previously described [29]. The sequences were submitted to GenBank under the accession nos. MZ254917–MZ254920.

## 2.8. Sequence analysis

Taxonomic affiliation of the isolates was determined based on the identity of their 16S rRNA gene sequences with those in the GenBank database and with the nearest type strains in EZBioCloud database as described in [29].

## 2.9. Phylogenetic tree construction

To construct a phylogenetic tree, sequences were aligned using ClustalW algorithms in the Geneious application (Geneious Prime 2021.1; https://www.geneious.com) [30]. Sequences of the type strains (T) were obtained from EZBioCloud with the accession numbers AE016877 (*Bacillus cereus* ATCC 14579), LASD01000006 (*Burkholderia contaminans* LMG 23361), CP022823 (*Klebsiella quasivariicola* KPN1705) and HQ888847 (*Providencia thailandensis* C1112). Phylogenetic tree of the aligned sequences was constructed using the maximum-likelihood method with Tamura–Nei distance algorithm in molecular evolutionary genetics analysis (MEGA) application according to a previous report [31].

## 2.10. Morphological and biochemical characterization

Characterization of the selected isolates by determination of colony morphology, biochemical and growth characteristics and fermentation of various carbohydrates were carried out as described previously [32,33].

## 2.11. Optimization of culture conditions

To determine optimum culture condition and hydrocarbon for biosurfactant production, the strains were grown for different incubation times (3–11 days), temperatures (25–50°C), pH (3–9) and hydrocarbon sources (kerosene, diesel, octane and soya bean) in Mckeen medium and the $E_{24}$ value at each was determined.

## 2.12. Extraction of biosurfactant

Extraction of biosurfactant was carried out as previously described [34]. Briefly, activated cultures were incubated at 37°C for 7 days at 150 r.p.m. Culture supernatant was collected by centrifugation at 5000 r.p.m. for 20 min at 4°C and pH was adjusted to 2 with 1 M $H_2SO_4$. Equal volume of chloroform–methanol mixture (2 : 1) was then added and shaken vigorously for 5 min and allowed to stand until phase separation. The bottom solvent phase was then removed by a separating funnel and the upper aqueous phase was collected. The partially purified biosurfactant was concentrated by evaporation and preserved at −20°C until analysed.

## 2.13. Characterization of biosurfactants

The chemical nature of the partially purified biosurfactants was determined by various biochemical examinations. Ninhydrin test was performed as reported by Feignier *et al.* [35], biuret test was performed according to Patoway *et al.* [36], Molisch's test was performed according to the method of Vanavil *et al.* [37] and thin layer chromatography according to Lamilla *et al.* [38].

## 2.14. Determination of antimicrobial activity

Antimicrobial activity of selected isolates was determined using partially purified biosurfactant against clinical and environmental bacteria of both Gram-positive and Gram-negative strains including *Bacillus cereus* (ATCC 14574), *Staphylococcus aureus* (ATCC 6538), *Pseudomonas aeruginosa* (ATCC 9027), *Salmonella typhi* (ATCC 14028), *Vibrio cholera* (ATCC 14035) and *Escherichia coli* (ATCC 25922) by disc diffusion method as described previously [39].

## 2.15. Determination of stability

The stability of biosurfactant was assessed from the determination of $E_{24}$ under various conditions such as temperature, pH and salinity. Thermal stability was estimated by placing at 4–121°C for 30 min followed by cooling to room temperature. pH stability was evaluated in the range of pH 1–10 adjusted with 1 N HCl or 1 N NaOH. Salinity was assessed using NaCl of 2–7% w/v.

## 2.16. Degradation of diesel oil by the selected strains

Degradation of diesel oil by the selected strains was measured by the gravimetric method described by Ganesh & Lin [40] in 100 ml minimal salt medium (MSM) containing 1.8 g $K_2HPO_4$, 4.0 g $NH_4Cl$, 0.2 g $MgSO_4 \cdot 7H_2O$, 0.1 g NaCl, 0.01 g $FeSO_4 \cdot 7H_2O$ per litre enriched with 2% (v/v) filter-sterilized crude oil as the carbon source cultured at 37°C for 7 days at 150 r.p.m. The residual oil was recovered using the solvent extraction method by adding dichloromethane to the media. The media/solvent mixture was decanted into a separating funnel, shaken well and the organic phase was drained into a previously weighed beaker. After evaporation of dichloromethane, the beaker was again weighed until a consistent weight was obtained. The difference between the two weights provided the weight of the residual oil. The same procedure was used for oil extraction from the negative control media maintained under the same conditions without any inoculation. Degradation of oil was calculated by the following formula:

$$\text{Oil degradation (\%)} = \left\{ \frac{(\text{weight of oil recovered from uninoculated media} - \text{weight of oil recovered from culture media})}{\text{weight of oil introduced}} \right\} \times 100.$$

# 3. Results

## 3.1. Selection and characterization of biosurfactant producing bacteria

Following enrichment in Mckeen media supplemented with 0.1% kerosene, 13 colonies that produced emulsified halos on hydrocarbon overlay agar were initially selected (table 2). Further screening based on the drop collapse assay and emulsification index ($E_{24}$; table 2) sorted out four isolates as efficient producers of biosurfactants (SD4, SD11, SD12, SD13). Although the hemolytic test, a method traditionally used in the screening, was performed, the technique has been reported not very reliable to detect biosurfactant production [41]. The selection criteria, therefore, relied principally upon the results of the drop collapse method and $E_{24}$. All the four selected isolates showed a vigorous collapse in the drop collapse test and an $E_{24} > 50\%$. The four isolates were taxonomically identified from their 16S rRNA gene-based analysis (figure 1a), and morphological and biochemical characteristics (table 3). The 16S rRNA gene sequence of the four strains showed maximum similarity to species of *Bacillus*, *Burkholderia*, *Providencia* and *Klebsiella*, respectively. This taxonomic affiliation was further supported by the phylogenetic relationship of the isolates with their closest type strains (figure 1b). Morphological and biochemical analysis (table 3) suggested that the isolates were non-motile, indole-negative and catalase-positive strains and, except the *Bacillus* strain (SD4), all were Gram-negative. All the four isolates could ferment glucose, fructose and sucrose. While the *Burkholderia* (SD11) and *Klebsiella* (SD13) strains also fermented raffinose, rhamnose, mannitol and lactose, *Bacillus* (SD4) and *Providencia* (SD12) isolates did not. Other biochemical properties such as cellular arrangement; citrate, nitrate, urease, methyl red, Voges–Proskauer, starch hydrolysis, deep glucose agar tests; and oxygen relationship were also determined (table 3) and found consistent with the taxonomic annotation according to Bergey's manual [42].

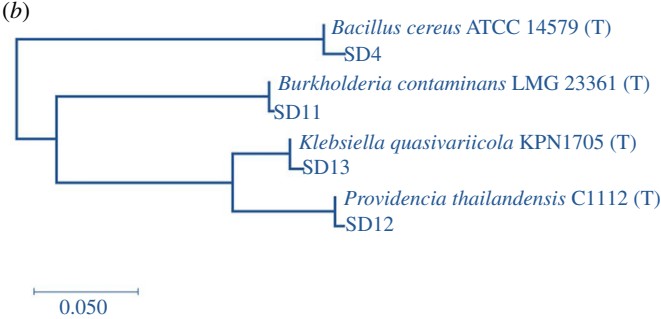

*(a)*

| isolates (accession no.) | BLAST result (top hit sp.) | percent identity |
|---|---|---|
| SD4 (MZ254917) | *Bacillus cereus* | 100 |
| SD11 (MZ254918) | *Burkholderia contaminans* | 100 |
| SD12 (MZ254919) | *Providencia stuartii* | 99.58 |
| SD13 (MZ254920) | *Klebsiella variicola* | 99.58 |

*(b)*

**Figure 1.** Phylotypes of the selected isolates. (*a*) Taxonomic affiliations of the four isolates based on sequence identity of their 16S rRNA genes. Accession numbers of the strains are provided in parentheses. (*b*) Phylogenetic tree of the isolates and their closest type strains (T).

**Table 2.** Screening of the isolates for biosurfactant production.

| isolates | hydrocarbon overlay agar[a] | drop collapse[b] | blood hemolysis[a] | emulsification index ($E_{24}$) (%) |
|---|---|---|---|---|
| SD1 | + | − | − | 0 |
| SD2 | + | − | − | 0 |
| SD3 | + | − | − | 5 |
| SD4 | + | +++ | + | 62.5 |
| SD5 | + | − | − | 6 |
| SD6 | + | ++ | − | 50 |
| SD7 | + | − | − | 50 |
| SD8 | + | ++ | + | 47.8 |
| SD9 | + | − | − | 25 |
| SD10 | + | − | − | 0 |
| SD11 | + | +++ | − | 55 |
| SD12 | + | +++ | + | 70 |
| SD13 | + | +++ | + | 74 |

[a] + = positive result; − = negative result.
[b] +++ = vigorous collapse; ++ = moderate collapse; + = scanty collapse.

## 3.2. Optimum growth conditions for biosurfactant production

The influence of various culture conditions on the production of biosurfactants was analysed and presented as a function of the emulsification index ($E_{24}$, %) (figure 2). While the production continued as long as day 11, the highest $E_{24}$ was obtained on day 5 in the *Bacillus* (SD4) and *Klebsiella* (SD13) strains, and on day 7 in *Burkholderia* (SD11) and *Providencia* (SD12) strains (figure 2*a*). For all isolates, the optimum production temperature was found at 37°C (figure 2*b*). Although considerable biosurfactant production was also observed below this temperature, the production sharply dropped at 45°C in all isolates. Acidic conditions, on the other hand, appeared to favour biosurfactant production in the isolates with the highest yield taking place at pH 3 (figure 2*c*). In fact, the $E_{24}$ was considerably high in most of the acidic range from pH 3 to 6 and decreased below or above this range

**Table 3.** Morphological, cultural and biochemical characteristics of the selected strains. + = positive result; − = negative result.

| features | *Bacillus* SD4 | *Burkholderia* SD11 | *Providencia* SD12 | *Klebsiella* SD13 |
|---|---|---|---|---|
| colony morphology | circular, raised, entire, smooth, off-white colour | circular, raised, entire, smooth, off-white colour | circular, raised, entire, smooth, off-white colour | circular, raised, entire, smooth, off-white colour |
| slant characteristics | effuse | filiform | arborescent | arborescent |
| Gram staining | + | − | − | − |
| motility test | − | − | − | − |
| cell arrangement | single | single | single | single |
| indole test | − | − | − | − |
| catalase test | + | + | + | + |
| citrate test | + | + | − | − |
| nitrate test | + | + | − | − |
| urease test | − | − | − | + |
| methyl red test | + | − | − | + |
| Voges–Proskauer test | + | − | − | + |
| deep glucose agar test | grow on the surface of medium | grow on the surface of medium | grow throughout medium | grow throughout medium |
| oxygen relationship | strictly aerobic | strictly aerobic | facultative anaerobic | facultative anaerobic |
| starch hydrolysis | + | − | + | − |
| fermentation of carbohydrates | | | | |
| glucose | + | + | + | + |
| fructose | + | + | + | + |
| dextrose | + | + | + | + |
| sucrose | + | + | + | + |
| maltose | + | + | − | + |
| raffinose | − | + | − | + |
| rhamnose | − | + | − | + |
| mannitol | − | + | − | + |
| lactose | − | + | − | + |
| starch | + | − | + | + |

although small emulsification was still observed at pH 2 and 9. With regards to the use of hydrocarbons in the media, kerosene was generally found most suitable for biosurfactant production followed by diesel and soya bean (figure 2*d*). By contrast, when octane was used as the carbon source, the emulsification capacity was very poor except for SD11 which showed a relatively better emulsification with octane. It appears that SD11 was the only isolate that could use all the hydrocarbon sources equally well for emulsification.

## 3.3. Chemical nature of the biosurfactants

The preliminary chemical structure of the biosurfactants was assessed from a series of biochemical reactions (table 4). Biosurfactants of *Bacillus* (SD4), *Burkholderia* (SD11) and *Providencia* (SD12) strains were found negative in the ninhydrin and biuret tests and in TLC sprayed with ninhydrin suggesting the absence of amino acids, but positive in Molisch test and TLC exposed to iodine vapour indicating the presence of carbohydrates and lipids. Taken together, biosurfactants of the three isolates seem glycolipid in nature. The *Klebsiella* (SD13) biosurfactant, however, showed quite a contrasting result in

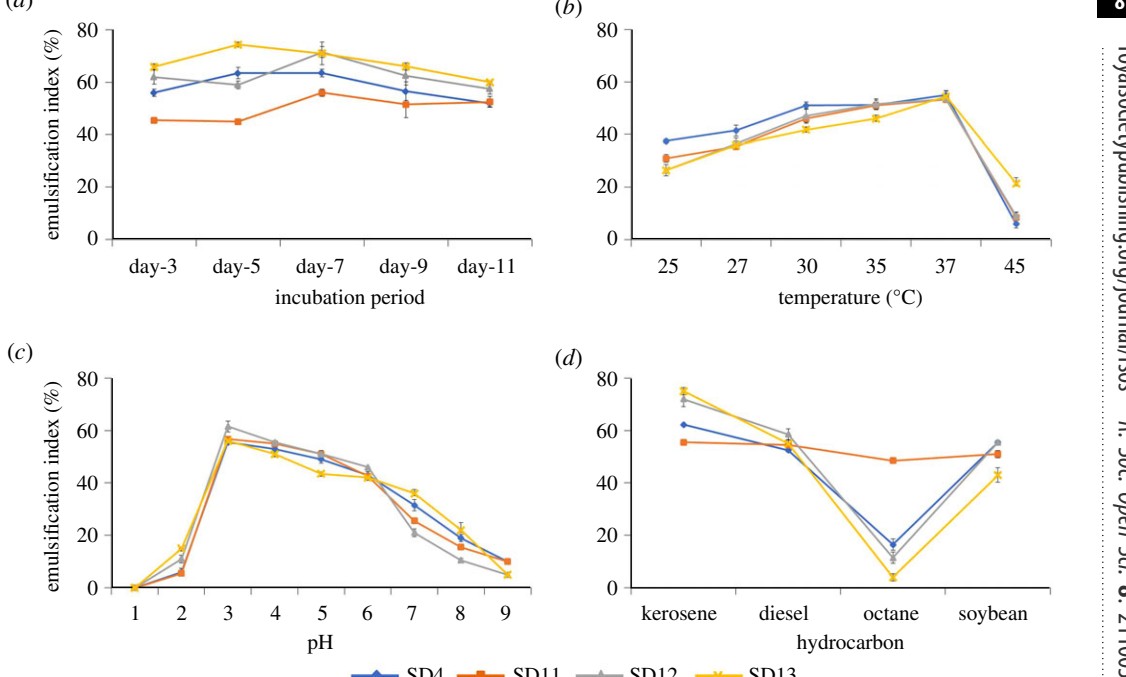

**Figure 2.** Effect of growth conditions on emulsification activity of culture filtrates. The isolates were grown at different sets of culture conditions such as incubation period (*a*), temperature (*b*), pH (*c*) and carbon source (*d*), and the emulsification index of culture supernatant was recorded. Error bars represent one standard deviation of the mean of three experiments.

**Table 4.** Results of biochemical tests for chemical characterization of the biosurfactants. $+$ = positive result; $-$ = negative result.

| | protein detection | | | carbohydrate detection | lipid detection | |
| --- | --- | --- | --- | --- | --- | --- |
| isolates | ninhydrin | biuret | TLC (ninhydrin) | Molisch | TLC (iodine vapour) | interpretation |
| SD4 | − | − | − | + | + | glycolipid |
| SD11 | − | − | − | + | + | glycolipid |
| SD12 | − | − | − | + | + | glycolipid |
| SD13 | + | + | + | − | + | lipopeptide |

the biochemical tests with negative reaction in the Molisch test and positive reactions in both protein and lipid detection tests suggesting it to be a lipopeptide.

## 3.4. Stability of the biosurfactants

The biosurfactant of each isolate was found to display a stable emulsification activity over a wide range of abiotic conditions (figure 3). Temperature sensitivity was assessed in a limit of 4–121°C. The *Klebsiella* (SD13) biosurfactant appeared most thermostable with relatively high emulsifying activity all along this temperature range (figure 3*a*). Biosurfactants of *Bacillus* (SD4) and *Burkholderia* (SD11) strains also showed similar stability with a slim decline over a temperature of 100°C. The *Providencia* (SD12) biosurfactant was, however, relatively less stable at temperatures below 25°C and above 70°C. High pH did not have much effect on the emulsifying capacity as the biosurfactants remained stable in both highly acidic and highly alkaline conditions (figure 3*b*). The biosurfactants of all four isolates maintained nearly constant values of $E_{24}$ over pH 2–9. The tolerance of the biosurfactants to ionic stress was also examined at 2–10% NaCl (figure 3*c*). The biosurfactants of *Burkholderia*, *Providencia* and *Klebsiella* strains showed similar emulsification activity forming stable emulsions at all these saline

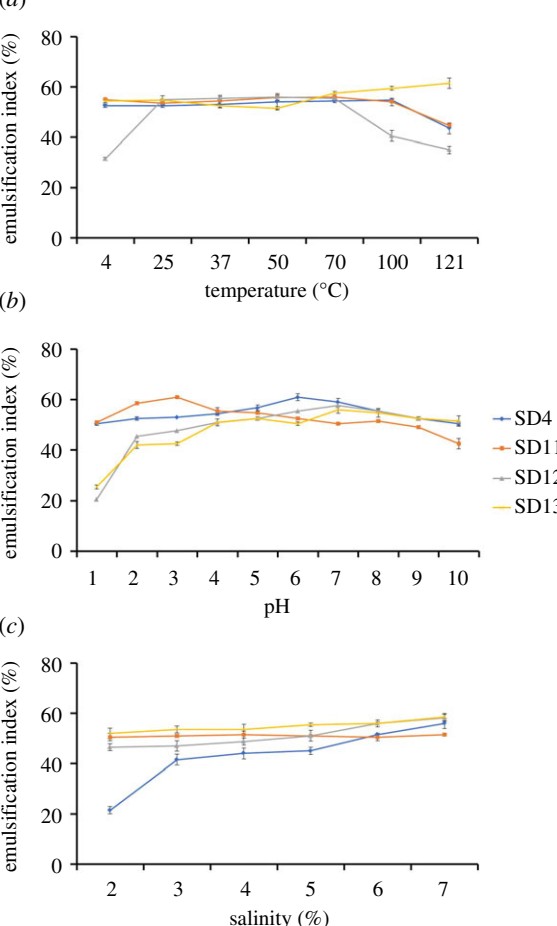

**Figure 3.** Stability of the biosurfactants at various (*a*) temperature, (*b*) pH and (*c*) salinity. The error bars represent one standard deviation of the mean, $n = 3$.

concentrations, whereas that of the *Bacillus* strain greatly diminished below 3% NaCl and gradually increased at higher ionic strengths.

## 3.5. Antimicrobial activity

The biosurfactants were tested as antimicrobial agents against six pathogenic or indicator organisms including *B. cereus*, *P. aeruginosa*, *S. aureus*, *S. typhi*, *V. cholera* and *E. coli* (table 5). The biosurfactant obtained from the *Bacillus* (SD4) strain was found most effective among the four isolates showing activity against a maximum of five test organisms having no effects against only *B. cereus*. *Bacillus cereus* was, in fact, the most unaffected organisms of the six test strains resisting biosurfactants of most isolates. Only the *Burkholderia* (SD11) strain exhibited antagonistic activity against it. *Vibrio cholera*, in contrast, was inhibited by biosurfactants of all four isolates. Biosurfactant from the *Providencia* (SD12) strain appeared the least effective with only two of the six test strains, *V. cholera* and *E. coli*, being inhibited. In general, the biosurfactants were more effective against the Gram-negative strains (69%) in comparison to the Gram-positive bacteria (38%).

## 3.6. Oil emulsification potential

The emulsifying capacity of the biosurfactants was measured using four different oil hydrocarbons, i.e. kerosene, diesel, octane and soya bean (figure 4*a*). Kerosene was found the most suitable substrate for emulsification followed by diesel and soya bean. Biosurfactants from all four isolates, especially *Klebsiella* ($E_{24} = 75\%$), demonstrated better emulsification with kerosene as compared to the other hydrocarbons. Considerable emulsion (approx. 50%) was also formed with diesel and soya bean. With octane, however, the emulsification activity was generally very poor with the exception of the *Burkholderia* (SD11) biosurfactant which showed relatively higher emulsification with octane. Although $E_{24}$ with octane was

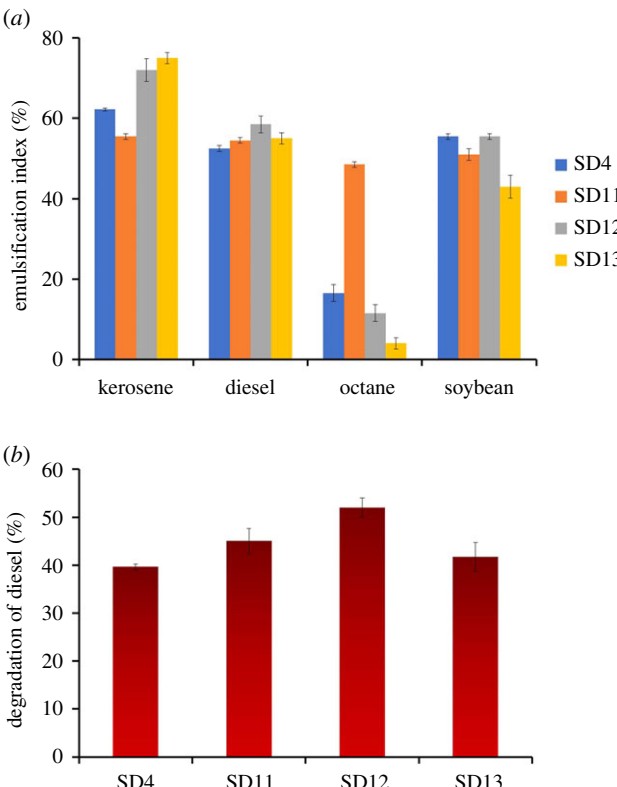

**Figure 4.** Emulsification activity of the biosurfactants on different types of oil (*a*), and degradation of diesel by the isolates (*b*). The error bars represent one standard deviation of the mean, *n* = 3.

**Table 5.** Antimicrobial activity of the biosurfactants against pathogenic or indicator bacterial strains. + = presence of activity; − = no activity.

| test strains | biosurfactant from | | | |
| --- | --- | --- | --- | --- |
| | SD4 | SD11 | SD12 | SD13 |
| Gram-positive strains | | | | |
| *Bacillus cereus* | − | + | − | − |
| *Staphylococcus aureus* | + | − | − | + |
| Gram-negative strains | | | | |
| *Pseudomonas aeruginosa* | + | − | − | + |
| *Salmonella typhi* | + | + | − | − |
| *Vibrio cholera* | + | + | + | + |
| *Escherichia coli* | + | − | + | + |

found to be 48.5% for the biosurfactant from *Burkholderia*, it was less than 20% for biosurfactants obtained from the other three isolates. Hence, the *Burkholderia* biosurfactant appears to have comparably broader substrate specificity. The ability of the isolates to degrade diesel oil was also studied by the determination of the amount of diesel oil left in the culture media after 7 days of growth (figure 4*b*); 40–52% degradation of diesel was achieved by the isolates which indicate their potential application in oil bioremediation.

## 4. Discussion

Bacterial surfactants play a major role in the emulsification of petroleum hydrocarbons; hence they are regarded as alternatives to chemical surfactants for superior properties like biodegradability, less

toxicity, eco-friendliness and high specificity [43]. The biosurfactant-producing bacteria are found in diverse environments but mostly isolated from places rich in organic hydrophobic contaminants [44]. In the present work, therefore, biosurfactant producers were searched in oil-contaminated sites. The isolates were examined by various screening methods including hydrocarbon overlay assay, blood agar assay, drop collapse method and emulsification indices since previous reports have recommended use of multiple techniques to screen for efficient biosurfactant-producing strains [44,45]. The initial screening was based on the hydrocarbon overlay assay sorting out 13 isolates for further selection. Blood agar assay, although widely used for screening of biosurfactant production [41], has been reported to give false positive and negative results [41]. Hence, the emulsification activity which is regarded to be a very reliable and accurate method to screen for biosurfactant production [43], together with the drop collapse assay, was basically considered [41] in this study in the final selection of four potent strains from the 13 isolates (46.15%). The four isolates were identified based on their 16S rRNA gene sequences. About 500 bp of the approximately 1500 bp sequence that has been found to be of high quality was used for the taxonomic identification. Limitations of identification by relatively short sequences were, however, described [46]. A nearly full-length sequence is said to be helpful for making a confident species or strain level identification [47], although several reports argued that a shorter sequence such as approximately 500 bp can also provide necessary divergence for the purpose [48,49]. In fact, both 500 and 1500 bp are common lengths to be sequenced and compared for phylotype determinations, and sequences of various lengths are found in databases and the literature [49–57]. Nevertheless, analysis of a nearly full-length sequence of the 16S rRNA gene is usually recommended, especially when reporting a new species or when it is necessary to differentiate between specific strains in a genus. Indeed, full-length sequences are supposed to provide relatively better resolution than short reads particularly for strains having high sequence similarity since it is indeterminate which segment of the 16S rRNA gene would provide the differentiation. On the other hand, for clinical isolates, the initial 500 bp has been reported sufficient for taxonomic differentiation [49]. Recently, Farrance and Hong examined 208 diverse bacterial sequences of 131 randomly selected genera by both the initial 500 bp and the 1500 bp sequences [48]. They found that 93.7% of the samples did not show any difference in the species level identification between the two approaches, whereas in only 5.3% of the samples the full-length sequences showed better resolution. Bacterial identification in the MicroSeq system is also based on 500 bp sequences, and identification using sequences shorter than 500 bp has been reported as well [58–62]. In the present study, analysis of the approximately 500 bp sequence of the four isolates exhibited greater than 99% identity to the closest GenBank sequences. Each was found to be affiliated with a different genus: *Bacillus*, *Burkholderia*, *Providencia* and *Klebsiella*. Bacterial strains from these four genera are well known as being capable of producing biosurfactants and degradation of petroleum hydrocarbons [43,44,63–65]. Most particularly, the members of the genus *Bacillus* have been very frequently isolated from the soil of oil-polluted sites and reported as an effective bioresource for biosurfactants [44,66]. The isolates, except the *Bacillus* strain, were Gram-negative. The dominance of Gram-negative species seems common in soil with a history of contamination by oil or its byproducts, a characteristic that has been suggested to contribute in the survival of these populations in such harsh environments [67].

In any strain, however, culture conditions play a major role in the growth of the strain itself and also in its production of a particular metabolite. It is, therefore, important to find out the optimum culture condition and suitable hydrocarbon source to achieve the maximum yield [68]. In the present work, the highest production of biosurfactant was found at day 5 or day 7 depending on the strain. A similar incubation time of maximum production was also reported in several other analyses [69–71]. Among the other factors, a temperature of 37°C and pH in the range of 3–6 appeared to be most suitable for the selected isolates to produce biosurfactants. The optimal temperature was close to the soil temperature (approx. 30°C) during isolation which indicates a direct correlation of biosurfactant production to the growth of the microbes under suitable temperature, i.e. higher production as the cell density increases. The optimal pH, in contrast, was found lower than that of the soil (approx. 8) from which the bacteria were obtained. While similar growth conditions were observed in several other bacteria, many species also showed optimum yield for different temperatures and pH, either higher or lower [45,68].

Preliminary characterization of the chemical nature of the partially purified biosurfactants indicated that the biosurfactants had glycolipid structures with the exception of the *Klebsiella* biosurfactant that was a lipopeptide. The glycolipid biosurfactants have recently gained special attention for their ecofriendly nature, high efficiency in biodegradation as well as other special activities such as pesticidal,

antifungal and antibacterial activities [72,73]. Accordingly, the glycolipid biosurfactants obtained in this work also showed antagonistic activity against several of the Gram-positive and Gram-negative bacteria. The most potent of these biosurfactants was that produced by the *Bacillus* strain (SD4) which demonstrated inhibitory effects against five of the six test organisms. Although previous research has shown that glycolipid biosurfactants such as mannosylerythritol have significant antimicrobial activity against Gram-positive bacteria [73], those extracted in the present study, in contrast, were usually more effective against the Gram-negative strains.

Another important feature of the glycolipid biosurfactants is their stability over an extreme range of pH, salinity and temperature [73] which is in line with the findings of the present research. Both glycolipid and lipopeptide biosurfactants of the present work exhibited good stability in maintaining emulsification at a wide range of pH, temperatures and salt concentrations thus indicating their suitability for application in extreme environmental or industrial conditions. The synthetic surfactants, on the other hand, are highly susceptible to such conditions. For example, salt concentrations over 2% NaCl were reported enough to inactivate a synthetic surfactant [74], whereas the emulsifying activity of the biosurfactants of this study remained unchanged from 2% to as high as 7% of NaCl.

To summarize, four bacterial strains and their secreted surfactants were characterized in this work. The partially purified biosurfactants had relatively high activity, formed stable emulsions with different hydrocarbons and showed good antimicrobial activity. Moreover, the biosurfactants also exhibited high levels of pH, salinity and thermal stability, and potential to degrade diesel oil, all which indicate their prospects for application in bioremediation and oil recovery processes under harsh conditions.

Data accessibility. Sequence data presented in this study can be found in NCBIData Bank with the accession nos. MZ254917–MZ254920.

Authors' contributions. F.A. and T.J.H. contributed to conception and design, and supervised the study; S.D. and F.A. carried out laboratory experiments and generated data; F.A., T.J.H. and T.D. analysed the data; S.I.C. and T.J.H. performed sequence and phylogenetic analysis; T.J.H. wrote and prepared the manuscript; T.D. prepared the electronic supplementary material, figure S1; S.A.Z., M.N.A.C., F.A. and T.D. helped in finding information regarding methods and discussion; M.S.U. reviewed the manuscript; all authors read and approved the final manuscript.

Competing interests. We declare we have no competing interests.

Funding. This project was partially funded by the University of Chittagong.

Acknowledgement. The authors would like to thank Dr Zobaidul Alam, Department of Microbiology, University of Chittagong for providing laboratory facilities and some chemicals. The authors also thank members of the Biochemistry and Pathogenesis of Microbes—BPM Research Group, Department of Biochemistry and Molecular Biology, University of Chittagong for their help.

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
