## [Peer Review File · Royal Society Open Science]

Review History

RSOS-211003.R0 (Original submission)

Review form: Reviewer 1 (nour el-gendy)

Is the manuscript scientifically sound in its present form?

Yes

Are the interpretations and conclusions justified by the results?

Yes

Is the language acceptable?

Yes

Do you have any ethical concerns with this paper?

No

Have you any concerns about statistical analyses in this paper?

Yes

Recommendation?

Major revision is needed (please make suggestions in comments)

Comments to the Author(s)

How did you extract diesel oil to follow up its degradation? add this to the manuscript.

Statistical and error analysis should be performed and added to the manuscript.

Accession number of all identified isolates on NCBI Gene Bank should be added to the manuscript.

Presumptive identification of the produced biosurfactant.

Review form: Reviewer 2

Is the manuscript scientifically sound in its present form?

No

Are the interpretations and conclusions justified by the results?

No

Is the language acceptable?

Yes

Do you have any ethical concerns with this paper?

No

Have you any concerns about statistical analyses in this paper?

No

Recommendation?

Major revision is needed (please make suggestions in comments)

Comments to the Author(s)

To the authors:

L57: Did you mean hydrophilic or hydrophobic?

L86: More information needs to be given about sample collection. What year were the samples collected? What was the gps coordinates of the sampling sites? How did you know that the sites were indeed oil contaminated? Were there any measurements on site or any recorded incidents of contamination?

L121: Information on the molecular methods needs to be expanded. How were the strains grown before DNA extraction and what was the method for DNA extraction?

L96: This part needs to be clarified. How was the initial colonies obtained? Did you first streak an aliquot of the suspension on a plate and then picked single colonies? Also, how did you ensure that the cultures were indeed pure cultures? Did you do multiple passages or obtain any microscopy pictures? If yes, any microscopy pictures should be disclosed.

L110: In the discussion section, you cite that blood agar test might not be very reliable. Is there a reason that blood assay needs to be included in this part?

L133: Proper citation of the program should be given (i.e. version, country where the program was patented)

L176: I think there are significant issues with this degradation assay. It doesn't confirm that the weight loss is specifically due to biological activity. There can be other abiotic factors involved, like evaporation. Did you at least use any controls that do not include cultures? Ideally, hydrocarbons need to be measured analytically and any metabolites detected are better proof of metabolic activity.

L226: As these isolates are from different species, would it be possible to indicate the species name before the strain name throughout the text? It would make it easier to follow.

Decision letter (RSOS-211003.R0)

Dear Dr Hossain,

The Editors assigned to your paper RSOS-211003 "Production Optimization, Stability, and Oil Emulsifying Potential of Biosurfactants from Selected Bacteria Isolated from Oil Contaminated Sites" have now received comments from reviewers and would like you to revise the paper in accordance with the reviewer comments and any comments from the Editors. Please note this decision does not guarantee eventual acceptance.

Please submit your revised manuscript and required files (see below) no later than 21 days from today's (ie 18-Aug-2021) date. Note: the ScholarOne system will 'lock' if submission of the revision is attempted 21 or more days after the deadline. If you do not think you will be able to meet this deadline please contact the editorial office immediately.

on behalf of Dr Ulas Tezel (Associate Editor) and Kevin Padian (Subject Editor)
openscience@royalsociety.org

Editor Comments to Author:

Thanks for your submission. As you will see, the referees offer some questions that need to be addressed in your revised version. Best wishes and please contact our editorial office if you need more time to revise.

Reviewer Comments to Author:

Reviewer: 1

Comments to the Author(s)

How did you extract diesel oil to follow up its degradation? add this to the manuscript.
Statistical and error analysis should be performed and added to the manuscript.
Accession number of all identified isolates on NCBI Gene Bank should be added to the manuscript.
Presumptive identification of the produced biosurfactant.

Reviewer: 2

Comments to the Author(s)

See attachment: "Review_letter_EE.pdf"

To the authors:

L57: Did you mean hydrophilic or hydrophobic?

L86: More information needs to be given about sample collection. What year were the samples collected? What was the gps coordinates of the sampling sites? How did you know that the sites were indeed oil contaminated? Were there any measurements on site or any recorded incidents of contamination?

L121: Information on the molecular methods needs to be expanded. How were the strains grown before DNA extraction and what was the method for DNA extraction?

L96: This part needs to be clarified. How was the initial colonies obtained? Did you first streak an aliquot of the suspension on a plate and then picked single colonies? Also, how did you ensure that the cultures were indeed pure cultures? Did you do multiple passages or obtain any microscopy pictures? If yes, any microscopy pictures should be disclosed.

L110: In the discussion section, you cite that blood agar test might not be very reliable. Is there a reason that blood assay needs to be included in this part?

L133: Proper citation of the program should be given (i.e. version, country where the program was patented)

L176: I think there are significant issues with this degradation assay. It doesn't confirm that the weight loss is specifically due to biological activity. There can be other abiotic factors involved,

like evaporation. Did you at least use any controls that do not include cultures? Ideally, hydrocarbons need to be measured analytically and any metabolites detected are better proof of metabolic activity.

L226: As these isolates are from different species, would it be possible to indicate the species name before the strain name throughout the text? It would make it easier to follow.

===PREPARING YOUR MANUSCRIPT===

===PREPARING YOUR REVISION IN SCHOLARONE===

Author's Response to Decision Letter for (RSOS-211003.R0)

See Appendix A.

Decision letter (RSOS-211003.R1)

Dear Dr Hossain

The Editors assigned to your paper RSOS-211003.R1 "Production Optimization, Stability, and Oil Emulsifying Potential of Biosurfactants from Selected Bacteria Isolated from Oil Contaminated Sites" have now received comments from reviewers and would like you to revise the paper in accordance with the reviewer comments and any comments from the Editors. Please note this decision does not guarantee eventual acceptance.

Please also ensure that the email address slamsumaiya93@yahoo.com is either updated to an active address or your colleague 'white lists' emails from the journal - at present, the address is not receiving messages from us, and we require all authors to have an active email able to receive messages from the journal.

Please submit your revised manuscript and required files (see below) no later than 21 days from today's (ie 02-Sep-2021) date. Note: the ScholarOne system will 'lock' if submission of the revision is attempted 21 or more days after the deadline. If you do not think you will be able to meet this deadline please contact the editorial office immediately.

on behalf of Dr Ulas Tezel (Associate Editor) and Kevin Padian (Subject Editor)
openscience@royalsociety.org

Associate Editor Comments to Author (Dr Ulas Tezel):
Associate Editor
Comments to the Author:
Manuscript RSOS-211003.R1

The responses of authors to the reviewers' questions/suggestions and the modifications done on the manuscript were evaluated critically by the editors. Here are some comments and suggestions to improve the manuscript before giving the final decision.

1. Authors responded Rev#1, Q2 on statistical and error analyses with a very generic description by just saying that they calculated mean and standard deviation of the data obtained from triplicate experiments in a new section. I suggest removing that very short section. Instead, authors may mention that in the captions of Figures 2, 3, and 4 where they should explain the data points and the error bars on that points. A text saying e.g. "error bars represent one standard deviation of the mean, $n=3$ ".
2. Remove the "grey background" from the figures and make data points and error bars more apparent.
3. Accession number of SD12 was misprinted
4. Figure 1a should contain the full species name that the alignment was made and the percent identity.
5. Although PCR was done with 27F and 1492R universal primers which would give an amplicon of c.a. 1400 bp, the sequences deposited on the GenBank were. Only around 500 bp length. Authors should explain the reason and discuss critically the drawbacks of species-level classification with that short sequences.
6. Figure S1 is missing.
7. Reference to Fig. S1 should be given as (see Supplemental Material, Figure S1)
8. Rev#2, Q3. The broth that the strains were grown before DNA extraction was still unknown.
9. Geneious software was not cited appropriately.

Editor Comments:

As you will see, there are some remaining issues to address in your manuscript that I hope you will find reasonable. All of these must be addressed satisfactorily or we regret that we will be unable to consider the manuscript further. Best wishes.

===PREPARING YOUR MANUSCRIPT===

Your revised paper should include the changes requested by the referees and Editors of your manuscript. You should provide two versions of this manuscript and both versions must be provided in an editable format:
one version identifying all the changes that have been made (for instance, in coloured highlight, in bold text, or tracked changes);
a 'clean' version of the new manuscript that incorporates the changes made, but does not highlight them. This version will be used for typesetting if your manuscript is accepted.

===PREPARING YOUR REVISION IN SCHOLARONE===

<https://royalsociety.org/journals/authors/author-guidelines/#data>. You should ensure that

you cite the dataset in your reference list. If you have deposited data etc in the Dryad repository, please include both the 'For publication' link and 'For review' link at this stage.

Author's Response to Decision Letter for (RSOS-211003.R1)

See Appendix B.

Decision letter (RSOS-211003.R2)

Dear Dr Hossain,

It is a pleasure to accept your manuscript entitled "Production Optimization, Stability, and Oil Emulsifying Potential of Biosurfactants from Selected Bacteria Isolated from Oil Contaminated Sites" in its current form for publication in Royal Society Open Science. The comments of the reviewer(s) who reviewed your manuscript are included at the foot of this letter.

on behalf of Dr Ulas Tezel (Associate Editor) and Kevin Padian (Subject Editor)
openscience@royalsociety.org

Appendix A

Dear Editors,

Thank you for giving us the opportunity to improve and resubmit our manuscript entitled "Production Optimization, Stability, and Oil Emulsifying Potential of Biosurfactants from Selected Bacteria Isolated from Oil Contaminated Sites". We believe that the revised manuscript satisfactorily addresses the Reviewers' questions and concerns. We would like to thank the Reviewers for their valuable comments that improved our manuscript. We hope it will now be suitable for publication in the *Royal Society Open Science*.

Thank you for your time and efforts on our manuscript. We look forward to hearing from you in due course and to respond to any further comments you may have.

Sincerely yours,

Tanim.

Tanim Jabid Hossain, PhD
Department of Biochemistry and Molecular Biology
University of Chittagong
Chittagong-4331, Bangladesh
Email: tanim.bmb@gmail.com

Response to Reviewers' Comments

We would like to thank the Reviewers for their thoughtful review of our manuscript and for the constructive remarks. We have carefully addressed the Reviewers' comments to improve and clarify the manuscript. Please find a detailed point-by-point response to all the comments below. Please note that the page and line numbers mentioned below refer to those in manuscript with track-changes.

Reviewer 1

1. How did you extract diesel oil to follow up its degradation? add this to the manuscript.

Response: Thank you for the important suggestion. For extraction of diesel oil, we used the method described by Ganesh and Lin (DOI: [10.5897/AJB09.811](https://doi.org/10.5897/AJB09.811)) that we cited in the submitted

manuscript. In the revised version we've described the extraction method in details (page 9, lines 184 – 191) as below.

“The residual oil was recovered using solvent extraction method by adding dichloromethane to the media. The media/solvent mixture was decanted into a separating funnel, shaken well and the organic phase was drained into a previously weighed beaker. After evaporation of dichloromethane, the beaker was again weighed until consistent weight was obtained. Difference between the two weights provided weight of the residual oil. Same procedure was used for oil-extraction from the negative control media maintained under the same conditions without any inoculation”.

2. Statistical and error analysis should be performed and added to the manuscript.

Response: We appreciate the Reviewer's suggestion. Accordingly, a new sub-section under the heading “Statistical analysis” has been added in the revised manuscript (page 9, lines 196 – 198):

“Statistical analysis

All experiments were performed three times and the data have been expressed as the mean \pm standard deviation.”

3. Accession number of all identified isolates on NCBI Gene Bank should be added to the manuscript.

Response: The accession number of the identified isolates has been provided in the Materials and Methods (page 6, lines 129 – 130) and in Figure 1.

4. Presumptive identification of the produced biosurfactant.

Response: Thank you for the question. We actually performed a few tests to identify biochemical nature of the biosurfactants (page 11, lines 241 – 250).

Reviewer 2

1. L57: Did you mean hydrophilic or hydrophobic?

Response: We thank the Reviewer for identifying this oversight. We've corrected the sentence as follows (page 3, line 57).

“Biochemically, the microbial surfactants are consisted of both hydrophilic and hydrophobic moieties [16]”

2. L86: More information needs to be given about sample collection. What year were the samples collected? What was the gps coordinates of the sampling sites? How did you know that the sites were indeed oil contaminated? Were there any measurements on site or any recorded incidents of contamination?

Response: We appreciate the Reviewer for pointing these out. We've now included the information about sampling time (page 5, line 87) and GPS coordinates of the sampling sites (Table 1 and supplementary figure S1). About the sites being oil-contaminated, in fact, oil leakage is very common in the country during regular activities at local fuel stations and the contaminated soil can be visible in naked eye.

3. L121: Information on the molecular methods needs to be expanded. How were the strains grown before DNA extraction and what was the method for DNA extraction?

Response: The necessary information about culture conditions and DNA extraction has been incorporated (page 6, lines 123 – 126) as follows.

“To amplify 16S rRNA gene sequences, activated cultures of the isolates were grown in nutrient broth at 37°C overnight and their genomic DNA was extracted using a Maxwell 16 Blood DNA Purification Kit (Promega, Madison, WI, United States) according to the manufacturer’s instructions.”

4. L96: This part needs to be clarified. How was the initial colonies obtained? Did you first streak an aliquot of the suspension on a plate and then picked single colonies? Also,

how did you ensure that the cultures were indeed pure cultures? Did you do multiple passages or obtain any microscopy pictures? If yes, any microscopy pictures should be disclosed.

Response: Thank you for pointing out the necessity to include this information. Pure colonies were obtained by multiple passages. Corresponding changes have been made to the revised manuscript (page 5, lines 96 – 99):

“After incubation at 37 °C for 3 days at 150 rpm, 100 µl of the suspension was spread over Mckeen agar plates and incubated at 37 °C. Single colonies from the plate were picked and repeatedly streaked on fresh plates until pure cultures appeared that were preserved as slant cultures.”

5. L110: In the discussion section, you cite that blood agar test might not be very reliable. Is there a reason that blood assay needs to be included in this part?

Response: Although blood agar test has been traditionally used for the detection of biosurfactants, it doesn't necessarily indicate biosurfactant activity. However, reason for us to perform this test is more as a biochemical property of the biosurfactants rather than as a screening technique.

6. L133: Proper citation of the program should be given (i.e. version, country where the program was patented)

Response: Thanks for pointing out the missed citation. Appropriate citation has now been given according to the developer of the program (page 7, line 137).

7. L176: I think there are significant issues with this degradation assay. It doesn't confirm that the weight loss is specifically due to biological activity. There can be other abiotic factors involved, like evaporation. Did you at least use any controls that do not include cultures? Ideally, hydrocarbons need to be measured analytically and any metabolites detected are better proof of metabolic activity.

Response: We appreciate the Reviewer for the insightful suggestions. We performed the degradation assay as described by Ganesh and Lin (DOI: 10.5897/AJB09.811). The same method has been frequently used in other works also (for example, Oluwaseun et al, 10.1016/j.scp.2017.07.001; Ma et al 10.1038/s41598-021-80991-5; Patowary et al, 10.3389/fmicb.2016.01092; Gontikaki et al, 10.1111/jam.14030 etc.). Yes, we used negative controls of uninoculated media. For better clarification, we've described the degradation assay in details in the revised version (page 9, lines 184 – 194) as below.

“Degradation of diesel oil by the selected strains was measured by gravimetric method described by [39] in 100 mL minimal salt medium (MSM) containing 1.8 g K₂HPO₄, 4.0 g NH₄Cl, 0.2 g MgSO₄.7H₂O, 0.1 g NaCl, 0.01 g FeSO₄. 7H₂O per liter enriched with 2% (v/v) filter-sterilized crude oil as the carbon source cultured at 37 °C for 7 days at 150 rpm. The residual oil was recovered using solvent extraction method by adding dichloromethane to the media. The media/solvent mixture was decanted into a separating funnel, shaken well and the organic phase was drained into a previously weighed beaker. After evaporation of dichloromethane, the beaker was again weighed until consistent weight was obtained. Difference between the two weights provided weight of the residual oil. Same procedure was used for oil-extraction from the negative control media maintained under the same conditions without any inoculation. Degradation of oil was calculated by the following formula:

$$\text{Oil degradation (\%)} = \left\{ \frac{(\text{Weight of oil recovered from uninoculated media} - \text{weight of oil recovered from culture media})}{\text{Weight of oil introduced}} \right\} \times 100$$

8. L226: As these isolates are from different species, would it be possible to indicate the species name before the strain name throughout the text? It would make it easier to follow.

Response: Thanks for the important suggestion. The species name before the strain name has been added throughout the text in the revised manuscript (page 10, lines 217 – 219, 227 – 228; page 11, lines 243 – 244, 247 – 248; page 12, lines 254, 256 – 257, 263, 265, 271, 274, 276; page 13, lines 284, 287 – 290).

Appendix B

Dear Editors,

We would like to thank you again for your precious time in reviewing our manuscript and providing the valuable comments. We have modified the manuscript according to the suggestions from Dr. Tezel and believe the manuscript has significantly improved with these changes. Please see our response to all the comments below.

We greatly appreciate your continued efforts on our manuscript and look forward to hearing from you in due course about the revised manuscript.

Sincerely yours,

Tanim.

Tanim Jabid Hossain, PhD
Department of Biochemistry and Molecular Biology
University of Chittagong
Chittagong-4331, Bangladesh
Email: tanim.bmb@gmail.com

Response to the Comments

We wish to express our appreciation to Dr. Tezel for the valuable suggestions. We agree with all his comments and revised our manuscript accordingly.

1. Authors responded Rev#1, Q2 on statistical and error analyses with a very generic description by just saying that they calculated mean and standard deviation of the data obtained from triplicate experiments in a new section. I suggest removing that very short section. Instead, authors may mention that in the captions of Figures 2, 3, and 4 where they should explain the data points and the error bars on that points. A text saying e.g. “error bars represent one standard deviation of the mean, $n=3$ ”.

Response: Thank you for the suggestion. The relevant section in the Materials and Methods has been removed and the information has been incorporated in the respective figure captions (page 9, lines 195-197; page 24, lines 624-630).

2. Remove the “grey background” from the figures and make data points and error bars more apparent.

Response: The figures have been accordingly revised.

3. Accession number of SD12 was misprinted

Response: We apologize for the typing error. The accession number of SD12 in Figure 1a has been corrected.

4. Figure 1a should contain the full species name that the alignment was made and the percent identity.

Response: Full species name and percent identity of the top hit species of BLAST result have been added in Figure 1a.

5. Although PCR was done with 27F and 1492R universal primers which would give an amplicon of c.a. 1400 bp, the sequences deposited on the GenBank were. Only around 500 bp length. Authors should explain the reason and discuss critically the drawbacks of species-level classification with that short sequences.

Response: Thank you for the valuable suggestion. We submitted partial 16S rRNA gene sequence that was of high quality. For technical reasons, peaks beyond this region were messy in the chromatogram. In the manuscript, we therefore described taxonomy at the genus level. Moreover, as suggested by Dr. Tezel, we also included a new discussion in the revised manuscript about identification using short sequence (page 14-15, lines 310-334) as follows.

“The four isolates were identified based on their 16S rRNA gene sequences. About 500 bp of the ~1500 bp sequence that has been found to be of high quality, was used for the taxonomic identification. Limitations of identification by relatively short sequences was, however, described [47]. A nearly full-length sequence is said to be helpful for making a confident species or strain level identification [48], although several reports argued that a shorter sequence such as ~500 bp can also provide necessary divergence for the purpose

[49,50]. In fact, both 500 and 1500 bp are common lengths to be sequenced and compared for phylotype determinations, and sequences of various lengths are found in databases and literatures [50–58]. Nevertheless, analysis of a nearly full-length sequence of the 16S rRNA gene is usually recommended, especially when reporting a new species or, when it is necessary to differentiate between specific strains in a genus. Indeed, full-length sequences are supposed to provide relatively better resolution than short reads particularly for strains having high sequence similarity since it is indeterminate which segment of the 16S rRNA gene would provide the differentiation. On the other hand, for clinical isolates, the initial 500 bp has been reported sufficient for taxonomic differentiation [50]. Recently, Christine and Sunhee examined 208 diverse bacterial sequences of 131 randomly selected genera by both the initial 500 bp and the 1500 bp sequences [49]. They found that 93.7% of the samples didn't show any difference in the species level identification between the two approaches, whereas in only 5.3% of the samples the full length sequences showed better resolution. Bacterial identification in the MicroSeq system is also based on 500 bp sequences, and identification using sequences shorter than 500 bp has been reported as well [59–63]. In the present study, analysis of the ~500 bp sequence of the four isolates exhibited >99% identity to the closest GenBank sequences. Each was found to be affiliated with a different genus: *Bacillus*, *Burkholderia*, *Providencia* and *Klebsiella*.”

6. Figure S1 is missing.

Response: We are sorry for the missing figure which we think might be because it was uploaded as a supplementary material. We will add this figure during the submission.

7. Reference to Fig. S1 should be given as (see Supplemental Material, Figure S1)

Response: Corrected accordingly (page 5, lines 86-87).

8. Rev#2, Q3. The broth that the strains were grown before DNA extraction was still unknown.

Response: Nutrient broth was used for both the pre-culture and activated culture. The sentence has been modified (page 6, lines 124-129) as follows.

“To amplify 16S rRNA gene sequences, cells from the stock culture were inoculated in nutrient broth containing 5.0 g peptone, 3.0 g yeast extract, 5.0 g NaCl in 1 liter distilled water and incubated overnight at 37°C. The activated cultures were further grown in nutrient broth at 37°C overnight and their genomic DNA was extracted using a Maxwell 16 Blood DNA Purification Kit (Promega, Madison, WI, United States) according to the manufacturer’s instructions.”

9. Geneious software was not cited appropriately.

Response: Thank you for pointing this out. For in-text citation, we followed the citation style suggested in the Geneious Prime website (<https://help.geneious.com/hc/en-us/articles/360044627352-How-do-I-cite-Geneious-or-Geneious-Prime-in-a-paper->). In the revised manuscript we’ve also cited a relevant paper (page 7, line 139; page 20, lines 490-492).

Thank you once again for your time and efforts on our manuscript. We hope it will now be suitable for publication in the *Royal Society Open Science*.